# Capabilities and Opportunities of Flexitarians to Become Food Innovators for a Healthy Planet: Two Explorative Studies

**Siet J. Sijtsema \***, **Hans Dagevos**, **Ghalia Nassar**, **Mariët van Haaster de Winter** and **Harriëtte M. Snoek**

Wageningen Economic Research, Wageningen University and Research, 6708 PB Wageningen, The Netherlands; hans.dagevos@wur.nl (H.D.); nassar.ghalia@gmail.com (G.N.); mariet.vanhaaster-dewinter@wur.nl (M.v.H.d.W.); harriette.snoek@wur.nl (H.M.S.)
**\*** Correspondence: siet.sijtsema@wur.nl

**Abstract:** To support the transition to a more plant-based diet, it is necessary to better understand flexitarians, i.e., individuals who curtail their meat intake by abstaining from eating meat occasionally without fully abandoning meat. Much of the research about eating (less) meat thus far has focused on motivations. However, a dietary shift toward less meat consumption also demands that capabilities and opportunities be taken into account. The present study explores the capability and opportunity variables in terms of enablers and barriers to reduced meat consumption. Focus group discussions (Study 1) and a survey study (Study 2) were conducted. Study 1 provides an overview of what food consumers perceive as capabilities and opportunities in the context of limiting meat consumption. Study 2 quantifies the aspects of capabilities and opportunities with a special focus on enabling and constraining aspects regarding plant-based meat substitutes. Both studies examine what Dutch flexitarians designate as capabilities and opportunities in transitioning to eating less meat in everyday life. More insight into this helps to find and facilitate food choices that make the flexitarian choice an easier and more obvious one and consequently contribute to flexitarians as food innovators for a healthy planet.

**Keywords:** flexitarianism; COM-B model; plant-based; meat substitutes; focus group discussion; food practices

## 1. Introduction

It has become abundantly clear over the past few decades that contemporary meat production levels and meat consumption habits are related to pressing environmental issues, human health problems, animal welfare issues, and global food insecurity. Consequently, a plethora of studies have highlighted the urgent need for a dietary transition toward less animal-based—particularly less (red and processed) meat-heavy—diets and more plant-based—vegetables, fruits, and pulses—eating patterns. This has led to the current situation in which broad consensus exists within scholarly circles concerning the sustainability and health benefits, as well as ethical merits, of a diet lower in animal-derived foods and higher in plant-based foods. Despite the value of a meat-reduced diet, this seems insufficient for many modern food consumers to limit their meat intake significantly and structurally. Although research has consistently found that environmental sustainability motives, health reasons, and animal welfare concerns are important to Western consumers in the context of meat reduction, in practice, however, it appears to be difficult to act accordingly.

From this perspective, a recent suggestion by Graça et al. [1] proved relevant and inspirational to the present study. Graça and colleagues argue that the increasing research interest in how to realize a dietary shift toward less meat-rich and more plant-based diets needs an overarching behavioral change model. Their proposal is to take the much-referred COM-B model by Michie et al. [2] into consideration as a helpful behavioral model—that includes motivation, opportunity, and capability—to improve our understanding of the difficulties and possibilities of transitioning toward lower levels of meat consumption.

Figure 1 depicts that the COM-B model consists of three components that influence behavior, influence each other, and are, in turn, influenced by behavior. This framework of the COM-B behavior system stands on the shoulders of the theory of planned behavior (TPB) by Ajzen and Fishbein [3] and is even more deeply indebted to the motivation–opportunity–ability (MOA) model by Ölander and Thøgersen [4]. The similarities between the MOA model and the COM-B model are striking. Both models encompass internal (psychological and physical) and external (environmental) key mechanisms for change: motivation and (cap)ability cover the first, and the opportunity factor covers the latter. Motivation entails the psychological reflective processes (e.g., planning, consequences, and beliefs) and automatic processes (e.g., emotional impulses and habits) that direct and empower a behavior. Capability is expressed as the individual's psychological (e.g., knowledge) and physical (e.g., skill) capacity to perform a given behavior. This relates to the concept of perceived behavioral control in TPB that concerns an individual's own perceptions of her or his (cap)ability to perform a given behavior. Opportunity refers to the external social (e.g., norms) and physical (e.g., environmental) factors in the individual's context domain that support or hinder a behavior. This component is recognizable from other behavior models in which the influence of the social and/or physical environment on the choices we are able or willing to make is referred to by notions such as facilitating conditions or external conditions.

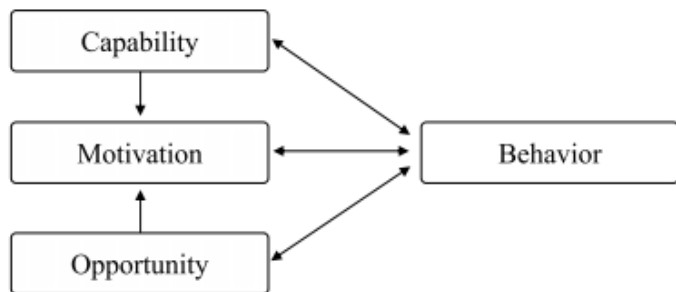

**Figure 1.** The COM-B model for understanding behavior change (source: Graça et al. [1]).

After bringing the COM-B model to the fore, Graça et al. [1] conducted a literature search on studies devoted to meat curtailment, meat substitution and adherence to plant-based diets. The focus of this review was on finding barriers (i.e., factors that hinder an individual from opting for a behavioral choice) and enablers (i.e., factors that are supportive of behavioral options) along the three components of the COM-B model. Graça et al. [1] found that, in the literature included, it is possible to detect several enablers with regard to opportunity and motivation and only a few for capabilities. In line with this, a smaller fraction of the relevant literature addressed variables in the domain of capabilities (6.4%) and opportunities (20%), while an overwhelming majority of studies addressed variables that were framed in the motivation domain (93.6%). In fact, this is unsurprising, given that a large majority of the studies included used a quantitative design—mostly survey studies. In such studies, much attention is traditionally given to motives, attitudes, and intentions. Overall, this review study by Graça and colleagues [1] taught that a "motivational perspective" often prevails in the field of reducing meat consumption and eating more plant-based diets. Consequently, variables connected to capability and opportunity have received much less attention thus far.

Therefore, taking the proposals and findings of the Graça study as our starting points, the present work aims to explore the domains of opportunity and capability a little further. In contrast to many survey studies found in the literature review by Graça et al. [1] and in an overview of current studies on meat eaters and meat reducers by Dagevos [5], our approach is, first of all, qualitative (Section 2). The second study reported in this article is quantitative, focusing on the opportunities and capabilities of meat-substitute consumption (Section 3). Studies with a quantitative design easily overlook the vital domains of opportunities and capabilities, which have remained under-researched as a result.

Even though the field of meat reduction and flexitarianism has blossomed in recent years [5], this does not imply that it has reached a stage that leaves room only for quantitative studies. Recent qualitative studies [6–11] prove that interview-based research produces insightful results, highlighting mundane barriers or ordinary facilitators in the process of lowering meat consumption. Such everyday difficulties or possibilities are frequently situated in the domains of capabilities and opportunities. In the closing section (Section 4), we will return to these issues in combination with the results obtained in the two explorative studies reported in the remainder. Both studies investigate what Dutch flexitarians perceive as capabilities and opportunities in transitioning to eating less meat in everyday life. More insight into this helps to answer the questions of whether and to what extent flexitarians could be referred to as food innovators for a healthy planet.

## 2. Study 1—Flexitarians in Focus Group Discussions

### 2.1. Introduction

Flexitarianism could be described as abstaining from eating meat on a part-time basis. A flexitarian, then, curtails her or his meat intake by abstaining from eating meat occasionally without fully abandoning meat. A flexitarian diet includes meat, in contrast to those of vegetarians who follow a meat-free diet and vegans who follow a strict plant-based diet from which all animal-based foods are excluded. In practice, flexitarians (or meat reducers, or "reducetarians") abstain from eating meat several days a week, indicate that their meat consumption has decreased and express intentions to lessen future meat intake, or declare that they hold positive attitudes toward a plant-based diet (see Dagevos [5] for an overview of definitions and studies on flexitarianism). Although flexitarians do not represent a uniform consumer segment per se, they have in common that they are motivated to moderate their meat intake to a certain degree. Although flexitarians' motives for adopting more meat-low habits could differ in sort and level, Graça et al. [1] rightly confirm that motivations behind eating less meat were more extensively studied recently in comparison to opportunities and capabilities. Both need further attention. Section 2 is devoted to this and presents the results of the in-depth discussions held with present-day flexitarians. Focus group discussions have focused on how contemporary food consumers are practicing their actual "flexitarian behavior", which hurdles they encounter, and how able they perceive themselves to be to execute this behavior. In doing so, this exploratory qualitative research intends to obtain a better impression of aspects that help or hinder flexitarians of today in performing their meat-reducing behavior.

### 2.2. Method

Four semi-structured focus group discussions with a total of 24 participants were held in Rotterdam in November 2019. Participants were recruited by a recruitment agency and were excluded if they consumed meat on a daily basis and did not partake in grocery shopping. The flexitarian participants were selected to represent a variation in terms of gender, age, education, and household composition resembling the general Dutch population. In return, they received monetary compensation for their participation. The number of participants per session was limited to six to create the right atmosphere for in-depth discussions. Sessions averaged approximately two hours of discussion and were led by one of the authors, who administered an open approach. All sessions were transcribed, and thematic coding was applied, including the key concepts from the COM-B model—motivations, capabilities, and opportunities—in terms of enablers and barriers. In addition, household management phases (e.g., planning, buying, storing, preparing, and eating) were coded. Finally, open coding was completed by relating to topics such as lifestyle, eating habits, nutrition, etc.

The discussion began with a general and personal introduction of the moderator and participants. After this, the flexitarian participants discussed the relative position of various protein types and shared their ideas on undertaking challenges of not consuming meat for half a year, which in turn supplied the essential barriers and enablers of opportunities and capabilities for the study. In addition, they were asked how to apply it in everyday life and

observe what types of changes they saw for themselves and in their environment when they started to eat less meat.

### 2.3. Capabilities and Barriers

During the group discussions, participants expressed several capabilities hindering lowering meat consumption, due to a lack of knowledge, skills, and inspiration. In each group discussion, the flexitarian participants mentioned that they experienced a shortage of information and knowledge concerning general health and nutritional values in terms of what their body needs, what should be compensated for when skipping meat during dinner multiple times a week, or when choosing a vegetarian lifestyle (see citations marked with A, B). In addition, they questioned the nutritional value of a vegetarian or vegan diet [A, C] and plant-based meat substitutes from the supermarket, more specifically. This appeared to be related to the lack of awareness about certain nutritional components, such as proteins, minerals, and vitamins. Participants felt insecure about all this and were uncertain about how much and how often they should eat certain nutrients, specifically flexitarians who intensively performed sports and wanted to stay energetic and in shape, as well as parents in their role as caretakers [B]. Next, a few participants expressed their general lack of trust in the information concerning nutrition, due to contradicting information [D]. For a few participants, the issue of trust also played a role in their perception of plant-based meat substitutes and their level/degree of processing. They assumed that much processing is needed to obtain a plant-based meat substitute with a taste experience similar to that of meat [E]. This seemed to tie into the overarching issue of processing food. In addition to experiencing an information shortage about a variety of issues, the participants also experienced a lack of skills in preparing a tasty meal without meat or a meal with a meat substitute. In particular, they addressed a lack of inspiration, such as having no ideas, difficulty with creative cooking, or no prior experience with unfamiliar ingredients [F]. Some participants mentioned that, in their perception, a meatless meal might include too many different and unfamiliar ingredients. These expressions showed that the lack of capabilities has an impact on the choice for meat in everyday life practices. The participants were raised with meat in their daily diet; therefore, they are more familiar with meat and its preparational and nutritional demands and not very knowledgeable or skilled concerning alternatives to meat, and consequently, they eat meat routinely [G].

*A. I think there is a lot in meat ... proteins, but also zinc and so ... whatever your body needs. ... I do think your body needs meat once or twice a week. ... I think you have to compensate a lot, if you want to get the rest of your vitamins and be completely vegetarian.*

*B. I find it difficult to determine, especially for my daughter, whether she is getting enough [nutrition]. ... I would love it if there were such an agency for food ... that gives some sort of guideline. If you are vegetarian, you should eat this during a day, and pay attention to that, or take more protein from this, or you can easily get it from that too. ... It is true that we all have to eat vegetarian, but how? And what? Give us some more guidelines.*

*C. What do we get from meat that we [when we eat vegetarian] need to get from other foods to stay healthy? I think that's information that a lot of people miss, and that's why sometimes ... a vegan diet really scares me, because I see that people get very sick. They just don't know how to deal with it in a healthy way. They go full on fruit and vegetables, develop thyroid problems. ... Your organs and body should continue to function properly.*

*D. Someone says yes, that's good, someone else says no, it's not that good because ... you get those kinds of health problems again, yes, then you don't know much yet. ... Contradicted [information] indeed then.*

*E. Thinking about vegetarian mince, it tastes the same as minced meat. Then an alarm bell rings for me, which makes me think, that's smart. When I take a bite, I think "oh, it's*

*chopped. It's vegetarian mince, but how?" Then that [processing] becomes more explicit for me.*

*F. In my opinion, in my experience, dishes like that . . . you have to put in so much work. Bringing in so much to make something out of it. If it has to have the same nutrients as what beef or chicken would give, I think it all looks nice, but I think it will take me 2 h and spend 20 euros for 1 dinner. It must also be easy to integrate . . . people don't have time every day to figure out what to eat, . . . it is just easy with meat.*

*G. [ . . . ] it sort of goes on autopilot. You eat what you always have eaten, what you . . . are raised with, without exploring other possibilities.*

### 2.4. Enabling Capabilities

Our flexitarian participants also mentioned several enablers that supported their consumption of plant-based meat substitutes. Information about diet in general and the nutritional value of meat substitutes was expressed to be beneficial for the participants. Moreover, the need for information about product preparation and the skills and equipment was addressed. With regard to both knowledge and skills, inspiration was needed, preferably for an easy-to-prepare and tasty meal (see citations marked with H, I). More specifically, participants found it encouraging to know what types of food products meat could be replaced with in the diet in general as well as in more familiar or even traditional dishes. Additionally, the convenient preparation of meat substitutes, especially for this type of product, a comparable preparation as meat, was thought to be helpful. A few participants addressed the issue of convenience from another point of view: in their perception, vegetarian or new dishes need more time; thus, they wanted sufficient time in their daily schedule to cook and try new recipes. Other participants appreciated meal solutions such as meal box schemes (e.g., HelloFresh) for easy preparation of well-explained convenient and tasty new vegetarian dishes [I]. Finally, participants mentioned that some vegetarian options of a traditional snack or dish were hard to distinguish from the meat version and that they liked such options for serving to meat-loving friends and family.

*H. Information about substitutes. And ideas [recipes] too. I also think about recipe reviews as an idea. That you have some opinions from people about substitutes. For example, that you can read that on the internet first, or on an app. . . . And also, nutritional information.*

*I. I am not a kitchen princess. I learned to cook from my mother and my grandmother. You have to learn it [vegetarian cooking] too. Someone has to show you. I think Hello Fresh is ideal. Once in a while I get a vegetarian dish. For example, chickpeas from the oven, falafel and something else, very tasty.*

### 2.5. Opportunities and Barriers

When participants discussed social and psychological contexts, they mentioned several barriers. First, they experienced a lack of alternatives available in supermarkets, restaurants, and canteens. Supermarket chains differed in the size of the assortments and the visibility of the plant-based alternatives (see citation marked with J). Participants mentioned that these products should have a more central position. Next, the menus in restaurants and canteens were not always found to be sufficient. Moreover, availability at home differed, for example, not having plant-based meat substitutes available in the fridge when starting cooking. For the out-of-home situation, some participants believed that the availability and variety of plant-based options on the menus of restaurants and fast-food chains should be improved [K]. In addition, advertisements supporting the purchase and consumption of meat were mentioned as barriers.

Next, there was the social context concerning lack of social support, which was represented in terms of unwillingness or reluctance from family and friends [L, M]. Participants stated that they faced difficulties in meat reduction when family members did not join them. One of the examples mentioned is that of a mother who has a son who wants to

eat meat, while her daughter does not want to eat meat. Additionally, the issue of social prejudice is addressed, and a few participants pointed out that, between friends, there is social pressure between meat lovers and vegetarians or vegans, especially when referring to a male group of friends [M]. Moreover, some participants recalled family traditions or values on special occasions, such as a celebration and serving food from the barbeque.

> *J. Now meat is the most eye-catching, all attention is paid to how it is presented in the supermarket. Vegetarian is often in a corner further away, or on the sidewall.*

> *K. [participant eats at work about 5 days a week] … how am I going to organize vegetarian food at work? It takes more time, more effort … I have the opportunity to get a pasta just around the corner … I am happy when I have the time to eat at all, at work.*

> *L. But when eating at friends … They have to take you into account. I wouldn't do it [ask for vegetarian dish] that quickly myself, I guess. Maybe once. But rather not …*

> *M. That vegetarian friend we have, who is part of a group of friends, altogether we are going to celebrate Christmas. He is the only one who does not like meat. All those boys are teasing him about not being able to eat anything.*

*2.6. Enabling Opportunities*

The flexitarian participants also experienced enablers from their social and physical contexts. They found it supportive to have sufficient alternatives to meat when eating both at home and out-of-home. Thus, a large assortment of meat substitutes and a central position in a supermarket were regarded as supportive, as well as more tasty meat-free options on the menus of restaurants in general and vegetarian restaurants specifically (see citations marked with N). For the home situation, it was also helpful to have the right ingredients available, sufficient cooking utensils, and time to prepare the vegetarian dish. In the supermarket, the position of plant-based meat substitutes, the role of advertisements, and discounts on meat substitutes were perceived to be beneficial to increasing consumption. For restaurants and canteens, ample tasty vegetarian varieties should be available.

To tackle the cultural element of family traditions and traditional foods or dishes, vegetarian options being as tasty as the traditional option were appreciated by some of the flexitarians [O], while others did not mind.

Participants found the willingness and social support of close ones to be an encouraging factor [P]. For example, family members, such as a daughter who had chosen to eat less meat, sparked a discussion with her family, inspiring them to lower their meat consumption as a family [Q]. This type of change consisted of talking about it, informing each other, trying new options, and providing each other with recipes. This meant that not only is the support factor important but also the involvement of other members and doing it together. Furthermore, significant others, such as doctors or famous athletes, were mentioned to influence dietary choices, as well as the position of schools teaching their children [Q]. This physical and social environment supported flexitarians to transform toward routines with more meat-free days [R].

> *N. It is also very much sorted by category in the supermarket. That I really think why isn't it mixed up. I also want that vegetarian chicken to be next to the real chicken. Give people the choice.*

> *O. For example, I was recently on a birthday. … there were bitterballen [snack] on the table. I had taken one, later I was told that it was a vegetarian one. But there was no difference in texture or taste or whatever … I would now easily get a vegetarian bag of bitterballen next time, because it tastes the same anyway.*

> *P. When you are a household with 2 or 3 persons, you have to decide with these guys, we are gone do this [eat less meat together].*

> *Q. My daughter tells us how to prepare it or gives tips. Start young. Especially if you do that in primary schools, they often take the information home.*

*R. I'm thinking about a new lifestyle. That you are used to no meat on Monday. Then you have a good product that you like on Tuesday . . . you have to get used to it and at some point you choose not to do it [eat meat].*

### 2.7. Interrelationship COM

During the coding process and checking the mentioned citations, it appeared that, from the perspective of everyday-life cooking and eating practices, there was—in accordance with the COM-B model (Figure 1)—an interrelationship between motivation, opportunity, and capability. The quotes below provide additional examples of the fact that the three component parts are often intimately connected. The first citation (marked with AA) shows the close relationship between habits and traditions (motivation) of eating meat and vegetables in the family, which resulted in the fact that flexitarians experienced a lack of information to change their meat-eating behavior (capability). The second example [BB] shows that there should be a balance between reason (motivation), assortment of alternatives (opportunity), knowing how to prepare (capability) tasty alternatives, and doing it together with the family (opportunity—social context). The third citation [CC] shows the influence of families (opportunity—social) of different generations; this man mentioned both his daughter and mother, as well as another relevant other: the doctor of the mother who made the participant realize his lack of knowledge about nutritional aspects of meat and meat substitutes. The traditional diet with meat and its associated dishes made the participating flexitarians feel more confident about what they got when eating meat, its satiety, and how much and how often they should eat meat. When changing this to eating meat less often, they were not sure about how to compensate and obtain the right nutrition and what replacements or substitutes to select from. In addition, they were unfamiliar with recipes and preparing vegetarian dishes because they had not grown up with the preparation of these dishes. These citations perfectly show the restrictions flexitarians experienced in practice and the need for the alignment of motivations, opportunities, and capabilities in everyday life.

*AA. Meat and vegetables . . . I was raised like that. Not really familiar with vegetarian. Grown up with that [traditional] kind of food. And now you're in a kind of a transition . . . There is often a lack of information. What should I do? It is quite a process before people are really used to it. That it becomes a habit.*

*BB. Combination of . . . know why you do it, and if you have good alternatives not to eat meat, and . . . easy to prepare recipes. . . . That you can easily achieve it without having to put in too much effort. Those tofu burgers that I made, I really had to add a lot . . . whole battery of herbs to give it a little flavor. If you can put something simple on the table, you just grab it and you're done. And you know what you are doing it for, and you are all doing it together, then I think that has a good chance of success.*

*CC. We also started to eat less meat at home. . . . Our youngest daughter has recently become vegetarian. . . . therefore, we are also becoming increasingly aware of this. We have consciously gone back a bit in meat, 2 or 3 times a week, and try carefully to have good replacements for it. Cheese every day, always on bread to work. Peanuts too. Especially lentils often as meat substitute. And what we do, . . . steak once a week, and next to it, for example, . . . a Italian burger is also a tasty meat substitute. Especially from the health perspective. You notice that you even start to like it [plant-based meat replacer] better, the more often you eat it. . . . then I was at my mother's house, she went to the doctor, who told her, you have to eat meat . . . I also started thinking, do I eat enough meat myself? . . . Ate meat twice [this week]. Isn't that actually too little? . . . Thus, I am conscious of eating less meat, but . . . I am not really looking for the right replacements, . . . I might have to put in a little more time for that. To find out carefully. . . .*

During the focus group discussions, several issues were discussed, of which we described those related to capabilities and opportunities. Strikingly enough, participants hardly referred spontaneously to motivational variables, such as the price or affordability

of vegetarian food or plant-based meat substitutes, or to environmental sustainability or animal welfare.

### 2.8. Wrap-Up Study 1

Our qualitative exploration deepens the insights on barriers and enablers of opportunities and capabilities experienced by flexitarians to consume less meat. As a consequence, our exploratory research adds to that of Graça et al. [1]. Table 1 summarizes the results obtained.

**Table 1.** Summary of barriers and enablers for capability and opportunity in meat reducing behavior, especially for the main meal. formatting code—bold: psychological; italic: physical; normal: social.

| Everyday Practices | Planning and Buying | Preparing | Eating | Other |
|---|---|---|---|---|
| **Capability** | | | | |
| **Barriers** | **Lack of knowledge: nutrition, preparation and recipes** | *Lack of skills; preparing vegetarian dish perceived as too difficult and with too many ingredients* | **(Childhood) memories of bad-tasting meat substitutes** | **Lack of trust in processed meat substitutes** |
| **Enablers** | **Information about nutrition, preparation and recipes** | *Convenient preparation Information about preparation Similar preparation to meat* | **Taste adoption of meat substitutes** | |
| **Opportunity** | | | | |
| **Barriers** | **Lack of options: accessibility and availability, lack of variety and visibility in supermarkets and restaurants** | *Lack of kitchen utensils Lack of time* | Lack of social support during meal | Social prejudice towards plant-based diet |
| **Enablers** | **Sufficient alternatives, big assortment at supermarket, in fridge at home and at restaurants** | *Sufficient equipment Sufficient time* | Social meal engagement | Social supportiveness and willingness |

## 3. Study 2—Flexitarians Inquired

### 3.1. Introduction

In line with Study 1, the aim of Study 2 is to provide further insight into the significance of capability and opportunity with respect to flexitarianism by focusing on the consumption of meat substitutes as a case study. Meat substitutes offer an alternative "meat-like" choice for flexitarians. As such, meat analogs facilitate meat reduction [12]. This is in accordance with the claims of participants in Study 1, who indicated that meat substitutes are helpful in transitioning to eating less meat. In the case of Study 2, a vegetarian burger patty was used specifically as an example. Hence, a cross-sectional study was administered, using a "within-subject" design to quantitatively explore the extent to which the capabilities and opportunities of meat reduction are applied to meat substitute consumption. The study was conducted through an administered online survey. In this exploratory survey, we look at enablers without barriers to eating meat substitutes to provide support for innovations and intervention strategies. We expect the main enablers mentioned for decreasing meat consumption to be akin to meat-substitute consumption, as shown by Study 1. Furthermore, Study 2 investigates the substitute consumption behavior of Dutch flexitarians and the motives behind their meat substitute choices.

### 3.2. Method

#### 3.2.1. Participants

The study pertained specifically to flexitarians, so a set of inclusion criteria was put in place. To be included in the target group, participants had to (i) not consume meat daily, (ii) live in the Netherlands, and (iii) do their own grocery shopping. The data collected for this study relied on convenience sampling and snowballing. This sample

is a subsample of another study conducted by Wageningen Economic Research in the Netherlands. A total of 685 respondents retrieved an online Qualtrics survey programmed by the researchers. Responses that were incomplete, deliberately "automatic," or too quick to be reliable (n = 202) were excluded. From the remaining 483 respondents, those who indicated they do not consume meat (n = 237) and do not live in the Netherlands (n= 8) were removed from further analyses. A total of 238 flexitarian participants were deemed valid for subsequent analyses.

### 3.2.2. Measures

All of the respondents answered a series of questions concerning their meat consumption frequency, consumption of alternative options, and meat-substitute consumption frequency. To measure the capabilities and opportunities of eating meat substitutes, capability and opportunity scales as enablers to start eating more plant-based meals were chosen from an original study by Graca et al. [13]. The items of these scales were adjusted to meat substitutes. Three items were used to measure capability (I know many recipes for meat substitutes; I know how to prepare meals with meat substitutes; I am able to judge the nutritional properties of meat substitutes). Four items were used to estimate opportunity (I have people close to me that support me (e.g., family, friends) in my consumption of meat substitutes; meat substitutes are accessible and convenient (e.g., supermarkets, restaurants); I know people who eat meat substitutes) and an additional homemade item (i.e., meat substitutes are tasty). The items were measured using a 7-point Likert scale (1 = Strongly disagree; 7 = Strongly agree). The capability and opportunity scales had Cronbach's alpha = 0.65 and 0.78, respectively.

An adaptation of the single-item food choice motives (FCM) scale by Onwezen et al. [14] was chosen to measure participants' choice motives concerning meat substitutes. We modified the word 'food' in the original statement to 'meat substitute' to comply with meat-substitute consumption rather than general food consumption: "It is important to me that the meat substitute (e.g., vegetarian burger) I eat on a typical day . . . " rated on a 7-point Likert type scale (1 = Not at all important; 7 = Extremely important). The items on the scale of Onwezen et al. covered various dimensions, including health, mood, convenience, sensory appeal, natural content, price, weight control, and familiarity. Additionally, three items were added to FCM: sustainable production and seasonal/local-based on the added value of sustainability motivations from Verain et al. [15] as well as safety, as discussed by Raaijmakers et al. [16]. The items were analyzed separately as single items instead of scales; thus, Cronbach's alpha was not calculated.

### 3.2.3. Sample

The 238 flexitarian participants (female = 186; male = 52) had an average age of 32 (SD = 10.73). Most participants (90%) had a higher-level education, and almost half had a lower-than-average annual income. Moreover, 78.9% of the participants consume meat three days or less per week, most of whom (29.4%) consume meat less than once a week. On average, the sample eat meat almost 2 days a week (M = 1.69, SD = 2.16). All sociodemographic variables are displayed in Table 2. A closer look at the meat substitute consumption frequency of flexitarians who eat meat substitutes shows that 65.5% (n = 156) of the total sample consumes meat substitutes (Figure 2). Predominantly, 39.7% of flexitarians consume meat substitutes 1–2 times a week, while 5.1% of flexitarians even eat meat substitutes almost daily.

To avoid any possible misunderstanding, it should be clear from the outset that this sample is neither in sociodemographic terms nor in meat substitute consumption patterns representative of the total Dutch population. An overrepresentation of female participants—especially the young and higher educated ones—means that we are dealing with a prominent consumer segment in adopting meat reducing diets. With respect to the consumption of meat substitutes, frequencies of respondents may be put into perspec-

tive by noting that currently, the Dutch consume per capita approximately 8 to 10 meat substitutes annually.

**Table 2.** Sample characteristics.

| Sample Characteristics | | |
|---|---|---|
| | **n** | **%** |
| *Gender* | | |
| Female | 186 | 78.2 |
| Male | 52 | 21.8 |
| *Age* | | |
| 18–34 | 172 | 72.3 |
| 35–60 | 62 | 26 |
| >60 | 4 | 1.7 |
| *Education level* | | |
| Less than high school | 2 | 0.8 |
| High school graduate | 21 | 8.8 |
| Bachelor's degree | 58 | 24.4 |
| Master's degree | 129 | 54.2 |
| Doctorate/Ph.D. or higher | 24 | 10.1 |
| Other | 4 | 1.7 |
| *Annual income* | | |
| Less than average | 118 | 49.6 |
| Approximately average (EUR 3000/month) | 52 | 21.8 |
| More than average | 68 | 28.6 |
| *Meat days per week* | | |
| Less than once | 70 | 29.4 |
| One | 37 | 15.5 |
| Two | 47 | 19.7 |
| Three | 34 | 14.3 |
| Four | 23 | 9.7 |
| Five | 18 | 7.6 |
| Six | 6 | 2.5 |
| Seven | 3 | 1.3 |

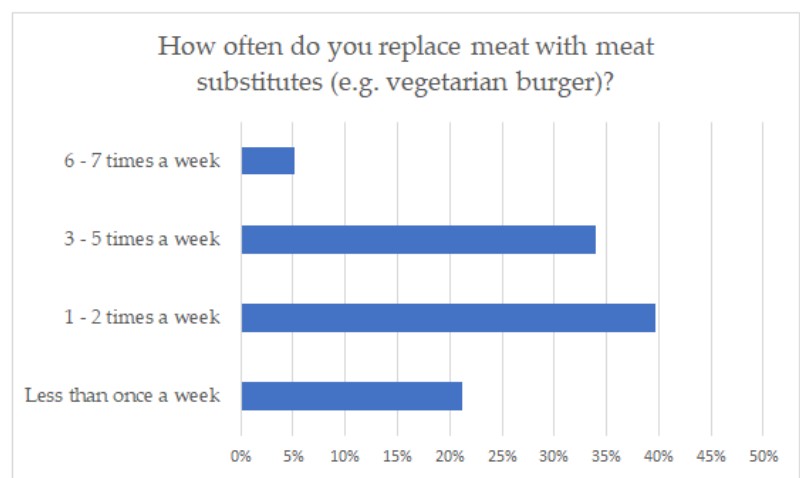

**Figure 2.** Meat substitute consumption (only those who indicated that they consume meat substitutes). N = 156.

*3.3. Results—Study 2*

3.3.1. Enabling Capability and Opportunity of Eating Meat Substitutes

The descriptive results of the enabling capabilities of consuming meat substitutes (Figure 3) showed that 66.5% of flexitarians in this sample were positive that they know many recipes for meat substitutes, while 25.8% indicated that they did not know many

recipes. The majority of the sample (86.1%) was confident that they knew how to prepare meals with meat substitutes. Finally, only 55.3% of the sample assessed that they could judge the nutritional properties of meat substitutes, while 32.8% stated that they could not.

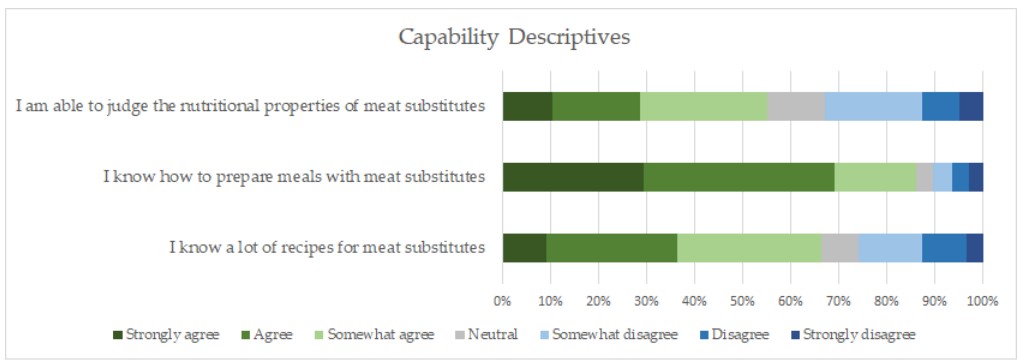

**Figure 3.** Enabling capabilities of eating meat substitutes.

Moving onto the opportunity enablers of consuming meat substitutes (Figure 4), the sample shows stronger agreement with the items with a high percentage of flexitarians (95.9%) who stated that they know individuals who eat meat substitutes. Furthermore, 83.3% of flexitarians agreed that meat substitutes are accessible and convenient in supermarkets and restaurants. Regarding whether flexitarians find meat substitutes tasty, 75.6% of the sample agreed that meat substitutes are tasty, while 15.4% did not agree with this statement, and others found it neutral (9.1%). Finally, 72.1% of the sample agreed that they had close friends or family that supported them in their meat substitute consumption, while 14% did not agree and 14% were neutral to this statement.

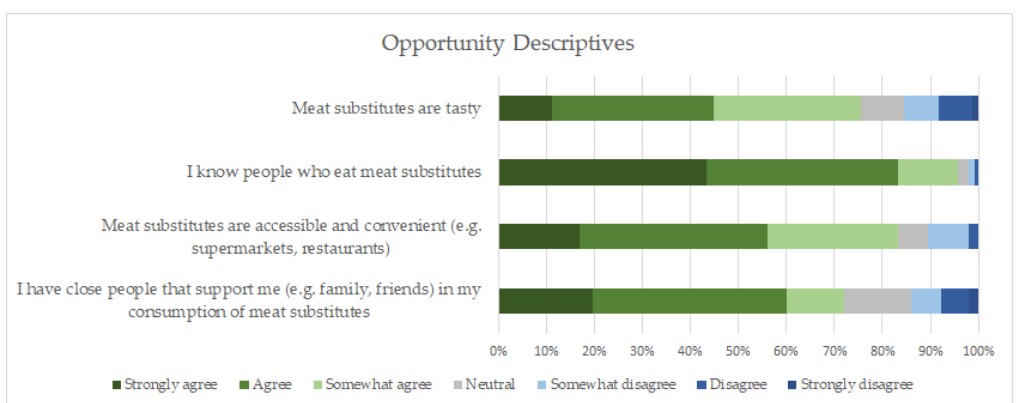

**Figure 4.** Enabling opportunities of eating meat substitutes.

The study also investigated the meat substitute choice motives represented in Figure 5. The results show that safety (M = 5.86, SD = 1.38) was ranked as the most important motive for eating meat substitutes. This was followed by the sensory appeal (M = 5.78, SD = 1.28) and health aspect (M = 5.66, SD = 1.20) of meat substitutes. Interestingly, sustainable production (M = 5.25, SD = 1.40) was a highly ranked motive for this flexitarian sample, whereas familiarity (M = 3.32, SD = 1.51) was one of the least important motives for meat-substitute consumption.

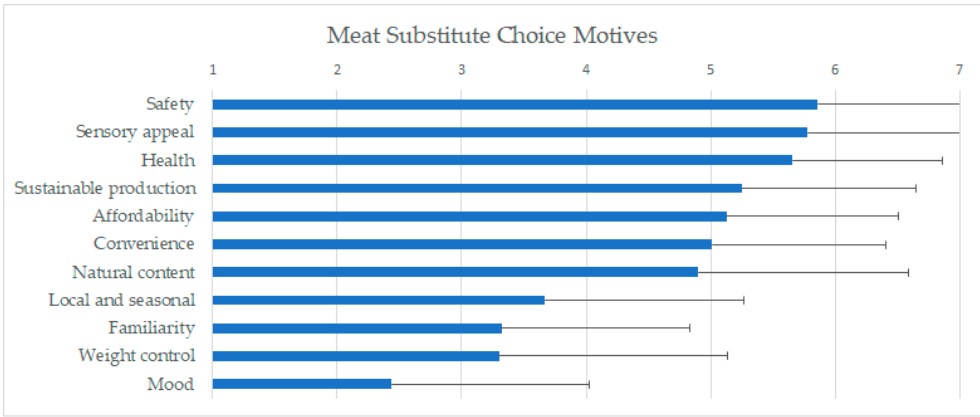

**Figure 5.** Meat-substitute choice motives. Replies to the question "It is important to me that the meat substitute (e.g., vegetarian burger) I eat on a typical day ... ".

### 3.3.2. Weighing Choice Motive, Capability, and Opportunity

The relation between the variables is best explained in Tables 3 and 4. Table 3 indicates the correlation between the capability and opportunity enablers for eating meat substitutes with the meat and meat-substitute consumption frequencies. Overall, the size of all significant correlations found was relatively low. As expected, meat-substitute consumption positively correlated with enabling capabilities and, to a lesser extent, with enabling opportunities for eating meat substitutes. For the items with the capability of eating meat substitutes, knowledge of recipes and meat-substitute preparation negatively correlated with meat consumption, indicating that the more participants knew of various recipes and meat-substitute preparation methods, the less likely they were to eat meat. For the enabling opportunity, the tastiness of meat substitutes and social support negatively correlated with meat consumption. This finding indicates that, the more participants find meat substitutes tasty and have a supportive social environment, the less likely they will be to eat meat. Thus, these two factors appear important in reducing the meat consumption behavior of flexitarians.

**Table 3.** Correlations between meat consumption frequency, meat-substitute consumption, capability, and opportunity (items).

|  | Meat-Substitute Consumption | Meat Consumption |
| --- | --- | --- |
| *Capability* (average of the three items) | 0.258 ** | −0.219 ** |
| I know many recipes for meat substitutes | 0.211 ** | −0.208 ** |
| I know how to prepare meals with meat substitutes | 0.239 ** | −0.182 ** |
| I am able to judge the nutritional properties of meat substitutes | −0.061 | 0.035 |
| *Opportunity* (average of the four items) | 0.183 * | −0.173 ** |
| I have people close to me that support me (e.g., family, friends) in my consumption of meat substitutes | 0.185 * | −0.220 ** |
| Meat substitutes are accessible and convenient (e.g., supermarkets, restaurants) | −0.021 | −0.011 |
| I know people who eat meat substitutes | 0.100 | −0.094 |
| Meat substitutes are tasty | 0.253 ** | −0.188 ** |

Significant at * $p < 0.05$ and ** $p < 0.01$.

The correlations shown in Table 4 provide insight into the association of capability and opportunity with meat-substitute choice motives. Evidently, the enabling capabilities for eating meat substitutes have a weak negative correlation with the familiarity of meat substitutes. This suggests that the more capable flexitarians are in consuming meat substitutes, the less that familiarity with meat substitutes is important to them. Interestingly, both capabilities and opportunities correlated positively with sustainable production of meat substitutes, but the correlation was low. This signifies that the more flexitarians value sustainably produced meat substitutes, the higher the capability and opportunity enablers for eating meat substitutes would be. Finally, sensory characteristics (e.g., texture,

appearance, smell, and taste) have a weak positive correlation with opportunities but no significant correlation with capabilities of eating meat substitutes.

**Table 4.** Correlations between capability, opportunity and meat-substitute choice motives [1].

|  | **Capability** | **Opportunity** |
|---|---|---|
| Opportunity | 0.625 ** | 1 |
| Health | 0.005 | 0.065 |
| Monitoring my mood | −0.056 | −0.033 |
| Convenience | −0.076 | 0.102 |
| Natural | 0.026 | 0.017 |
| Affordable | −0.040 | 0.034 |
| Weight control | −0.113 | −0.082 |
| Familiarity | −0.212 ** | −0.118 |
| Sustainable production | 0.198 ** | 0.212 ** |
| Local and seasonal | 0.078 | −0.019 |
| Safety | 0.006 | 0.086 |
| Pleasurable sensations | 0.027 | 0.193 ** |

[1] Respondents were asked how important the food choice motives were for their choice of meat substitutes. Significant at ** $p < 0.01$.

### 3.4. Wrap-Up Study 2

The findings revealed the importance of capability and opportunity in understanding meat-substitute consumption. Overall, capability and opportunity are reinforced with increased meat-substitute consumption and decreased meat consumption. In contrast to the participants of Study 1, who appear to be rather "light" flexitarians, eating a plant-based dinner only once or twice a week, the majority of the respondents of Study 2 are young highly educated females having a "heavy" flexitarian diet. The sample of Study 2 comprised a high number (65%) of so-called "heavy" flexitarians: those who consume meat two days a week or less (see [5,17]). This is an important aspect to bear in mind while reflecting on the results, as a high majority of flexitarians in this sample are more knowledgeable of preparing low-to-no-meat meals and are familiar with meat substitutes. Nevertheless, almost one-third of the "heavy" flexitarians still perceive difficulties with capability and opportunity in regard to meat substitutes. Only a small majority of the respondents reported that they were able to judge the nutritional properties of meat substitutes (capability). In other words, even experienced flexitarians experience a lack of nutritional knowledge about meat substitutes. Overall, Study 2 shows that most difficulties are in judging nutritional values (capability) and tasty alternatives (opportunity), particularly the latter, as sensory appeal (appearance, taste, and texture) of plant-based meat alternatives has been a well-known variable in studies on consumer acceptance and adoption of meat substitutes since the beginning of research in this domain; see [18].

With respect to external social and physical factors of opportunity, Study 2 revealed that a large majority of the respondents acknowledged the positive influence of people (close to them) who are in favor of eating meat substitutes as well as—to a lesser extent—of the availability and accessibility of meat substitutes in supermarkets.

### 4. General Discussion

Both studies that were reported in the present work explore understudied capabilities and opportunities for flexitarians to consume less meat and more plant-based meat alternatives in an everyday context. In Study 1, we obtained more detailed insights into barriers and enablers of several capability and opportunity variables. In Study 2, we showed that higher meat-substitute consumption and lower meat consumption frequency were related to enabling capabilities and opportunities. Simultaneously, flexitarians still experience barriers, both in social and physical contexts (opportunity) and in nutritional knowledge, cooking skills, inspiration or recipes (capability).

This article is inspired by Graça and colleagues [1], who justly noticed that meat reduction is not solely a matter of individual motivation. They call for taking capability and

opportunity variables into account in regard to a dietary shift towards meat reduction and plant-based eating patterns. Food consumption choices are deeply influenced by social and contextual circumstances and are not solely determined by consumer motives—ranging from environmental sustainability or animal welfare concerns to health-related and price-oriented motivations—but motives are also shaped and constrained by characteristics of the social and physical environment. Although we may have long known that this also holds for meat reduction (e.g., [19]), it appears that we have not given this sufficient attention in recent years. In essence, the latter also applies to qualitative research in the field of meat consumption and meat reduction. In effect, a research tradition that has a starting point with an interview-based study by Holm and Mølm in 2000 [20] has received little imitation in recent decades in comparison to the number of quantitative studies on flexitarianism [5].

In bringing this article to a close, in the following subsections, we briefly discuss how the current outcomes of capabilities and opportunities are related to previous research, and we highlight the need to maintain this "capability and opportunity perspective" in future research within the field of flexitarianism.

### 4.1. Capabilities

First, regarding capability variables, our two exploratory studies have pointed to the need of contemporary flexitarians for nutritional information. Both in Studies 1 and 2, consumers emphasized the lack of knowledge and uncertainty of nutritional aspects concerning nutrients in food products and diet in general but also specifically regarding plant-based diets and dishes. In Study 1, nutritional knowledge about meat diets and plant-based meat alternatives was related not only to the participants' own consumption patterns, but also to raising children or maintaining an active lifestyle. It was suggested that an information campaign about the adverse (health) impact of meat overconsumption or nutritional guidelines for vegetarian and flexitarian diets and dishes would be helpful as well as a comparison between the nutritional value of meat and its plant-based substitute on the package. In Study 2, it turned out that even "heavy" (female) flexitarians indicated not having enough nutritional knowledge—which is not very hopeful in view of the level of information of more meat-attached (male) consumers. Such results suggest that flexitarians' capacities could be increased and encouraged by nutritional knowledge on protein needs and information about nutritional value and satiating value through better displaying on product packages and/or public campaigns under the authority of trusted nutrition centers or non-governmental organizations.

In addition to this, the nutritional value of foods and knowledge about nutritious food is closely related to the healthiness of food. The "health issue" appeared to be an important issue in the context of both the healthiness of eating meat and the healthiness of meat reduction. As a result, it is unsurprising that health is known not only as an important motive for food consumers but also as a complicated motive in its consequences. That is, also in the realm of meat reduction, health has appeared to be a motive for lowering one's meat intake, e.g., [21,22], while it was also found that food consumers—including flexitarian segments—are not always convinced of the positive health effects of meat reduction, e.g., [17], or take the perceived health benefits of meat as a (compelling) reason to keep on consuming meat, e.g., [23].

Within the domain of capability, cooking skills was another prominent variable in the studies conducted. Participants in Study 1 referred to lack of cooking skills as a barrier for preparing a vegetarian dish. Other impediments mentioned were that plant-based dishes were regarded as more expensive, less tasty, or in need much more diverse types of ingredients or unfamiliar ingredients. Both studies also revealed that flexitarians kept on using meat as their baseline comparison. Their cooking skills as well as their social and physical environment were still largely based on meat-eating diets, habits, and traditions; for example, preparation was considered easier when vegetarian dishes were similar to meat. Although those who consume meat-free main meals more often reported a higher level of knowledge about eating low-to-no-meat dishes, even flexitarians who were

already shifting to less meat-rich diets indicated that they need support in their skills and knowledge to succeed in a flexitarian food style diet.

More generally, "capability building" is mostly about convenience; it is about being and becoming more certain and comfortable regarding more plant-based eating conventions and "mealing" practices. Putting more emphasis on capabilities in meat reduction research highlights how and to what extent a flexitarian diet fits within existing knowledge, interests, skills and (eating) routines.

### 4.2. Opportunities

Among other studies, recent qualitative research, e.g., [6,9,10], stresses the impact of the social context as a barrier to reducing meat consumption. The current study confirmed this. A flexitarian diet is sometimes hindered when, for instance, flexitarians do not feel comfortable preparing a meatless dish when friends are invited for dinner. Alternatively, consumers feel encouraged to reduce their meat intake when significant others are in favor of eating meat substitutes. An enabling social environment and favorable food consumer culture at large will be unmistakably important for the flourishing of flexitarianism in the foreseeable future. A focus on opportunities revealed that meat reduction, rather than being an individual issue, is a joint activity within a household, among family members or a group of friends. Through the lens of social context, it is noticed that it matters what children are taught at school, which food conventions are established, how appropriate meat products, meat substitutes or other alternative proteins are perceived in different situations (see [24]), or how traditions are transitioned by other significant factors, such as movie stars, influencers, athletes or doctors. Thus, in addition to personal motivations and capabilities, the social context is vital to promoting eating less meat by doing it together and being conscious of the social system and ("meaty") food cultural traditions.

Next to socio-cultural situations, and in addition to the work of Graça et al. [1], the present work also addressed the (limited) availability of plant-based foods in supermarkets or on the menus of restaurants as mundane opportunity variables. Additionally, the perceived sensory quality of meat alternatives remained a consumer issue. Study 2, for instance, showed that even "heavy" flexitarians pointed to taste as a barrier. Further development of tasty plant-based products and the availability and attractive presentation of a varied assortment in supermarkets and out-of-home outlets are necessary pathways to help the transition toward more plant-based eating habits.

Having said this, it is important to realize that the availability of meat substitutes is a necessary condition in the opportunity domain. In the Dutch food market, this precondition is met by a growing amount and assortment of plant-based meat alternative options on supermarket shelves and restaurant menus. The finding that Dutch flexitarian respondents still perceive availability as limiting their opportunity may be put into perspective if we consider that meat substitutes in many other countries are hardly or not available. This leads us to state explicitly at the end of this study that a flexitarian diet does not equal consuming meat substitutes alone. Although considerable attention was paid to processed meat substitutes, this does not imply that flexitarianism is narrowed down to the adoption of meat substitutes. On the contrary, a flexitarian diet definitely also means substituting meat with (unprocessed) vegetables and fruits, nuts, legumes and seaweed, or mushrooms and tofu.

### 4.3. In Closing

Our studies on flexitarians demonstrated that, in everyday life, the choice to eat less meat is an issue not only of personal motivation, but also of enabling capabilities, such as knowledge and skills, as well as of supportive opportunities provided by the social and physical environment. Several capability and opportunity variables were identified that both enable and constrain meat-reducing behavior. A "capability and opportunity perspective" pays special attention to everyday practices performed by individuals and everyday aspects that help or hinder flexitarian food choices, such as nutritional knowledge,

availability and tastiness of meat substitutes, unfamiliarity with plant-based meals, taste and convenience perceptions, uncertainty about cooking skills or reactions from relatives.

It was also emphasized that more research interest in capability and opportunity variables may be beneficial to studies on flexitarianism from a "motivational perspective" because it not merely adds to this approach, but also interrelates with it: in the COM-B system, motivations, capabilities and opportunities are intertwined [1,2] (see also [25]). In other words, gaining further insights into the capabilities and opportunities of flexitarians will generate further understanding of contemporary food consumers' underlying motivations, and vice versa. Therefore, finding and facilitating capabilities and opportunities for flexitarian consumer choices is critically important to help flexitarians become food innovators for a healthy planet.

**Author Contributions:** Conceptualization, S.J.S. and H.D.; methodology, S.J.S., M.v.H.d.W. and G.N., H.M.S.; formal analysis and investigation Study 1, S.J.S. and M.v.H.d.W.; formal analysis and investigation Study 2, G.N. and H.M.S.; writing—original draft preparation, S.J.S., H.D. and G.N.; writing—review and editing, S.J.S., H.D., G.N., H.M.S. and M.v.H.d.W. All authors have read and agreed to the published version of the manuscript.

**Funding:** This research was funded by the Investment theme "Protein Transition" of Wageningen University & Research. Next, it is part of the project Reverse Engineering of "KB Healthy and Safe food", funded by the Dutch Ministry of Agriculture, Nature and Food Quality.

**Institutional Review Board Statement:** Ethical review and approval were waived for this study, due to no biological tissue sampling.

**Informed Consent Statement:** Informed consent was obtained from all subjects involved in the study.

**Data Availability Statement:** The data presented in this study are available on request from the corresponding author. The data are not publicly available due to privacy.

**Conflicts of Interest:** The authors declare no conflict of interest.

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
