# Peer review of "Capabilities and Opportunities of Flexitarians to Become Food Innovators for a Healthy Planet: Two Explorative Studies"

_sustainability, doi:10.3390/su132011135_

Round 1

Reviewer 1 Report

The article was prepared based on the current literature on the analyzed issue. A broad analysis of the current scientific achievements concerning the analyzed topic was also carried out. The good foundations allowed for the proper selection of issues for the conducted research as well as the proper selection of research methods. The use of two research methods - focus research (as preliminary research, directing further, wider research) and questionnaire research gave very good results and is very appropriate from the point of view of the methodology of scientific research. The article is very well prepared. I have no critical comments. Probably not the entirety of the research has been presented, but from the point of view of the assumed goal, the number of research results and conclusions presented are sufficient. 

Author Response

Response to reviewer 1.

The article was prepared based on the current literature on the analyzed issue. A broad analysis of the current scientific achievements concerning the analyzed topic was also carried out. The good foundations allowed for the proper selection of issues for the conducted research as well as the proper selection of research methods. The use of two research methods - focus research (as preliminary research, directing further, wider research) and questionnaire research gave very good results and is very appropriate from the point of view of the methodology of scientific research. The article is very well prepared. I have no critical comments. Probably not the entirety of the research has been presented, but from the point of view of the assumed goal, the number of research results and conclusions presented are sufficient. 

Thank you for reviewing our work. We really appreciate your kind words and reflections.

Reviewer 2 Report

·The abstract needs to be written in more detail and depth

·More explanation of mathematical methods is needed

·Insufficient number of references

Author Response

Rebuttal guest Editors Dora Marinova and Diana Bogueva and below the response to reviewer 2.

Review and respons Diana Bogueva

First, I will introduce the COM-B model. What is it and then how could be used for understanding behaviour change? It was kind of abruptly introduced and if a person reading it is not familiar with it, that it is a model that proposes that there are three components to any behaviour - Capability, Opportunity, and Motivation and that based on the interaction any interventions must target one or more of these components to deliver and maintain effective behaviour change.  

Well, actually we think we’ve introduced the model efficiently and even put it in a ‘historical’ context of TPB and MOA. It is also clear that readers who want to know more could go to the Graca-paper (in which the introduction is neither very long by the way) or to the original Michie-work. For understanding the present work we believe we’ve provided the information needed.

In figure 1 the components are not presented in the right order. Not sure if this is intentional.  

We copied the figure from the Graca-paper and this way of presenting is very common (check Google images for instance)

I also wasn’t sure of what the letters meant [A, B, C, AA, BB,  etc.] at the beginning and then realised that they were related to the 24 participants. Maybe if there is a sentence, explaining this, I think it will leave the reader not wondering that.

See response Dora #1

I found the idea of [B] for “agency for food “ to provide “guidelines’ rather brilliant to be left without exploring it further.  I think it could be looking at in line with existing dietary guidelines in the Netherlands, maybe in conjunction with Eat Lancet recommendations, just a thought.

As in many other countries, the Netherlands has such an agency (Nutrition Centre) – but the respondent doesn’t seem to be aware of this. Anyway, inspired by your comment we’ve added some new text (see lines 614-15) 

Study 2 – it is not clear who administered the survey: “conducted through an administered online survey”. Is this the same company that helped with conducting the focus groups?

I was surprised to see the very high percentage of non-meat eaters participants: ”From the remaining 483 respondents, those who indicated they do not consume meat 415 (n=238)…. were removed from further analyses.” Maybe should add what is the percentage of vegetarians/vegans in the Netherlands, because it is a very high statistical representation. I am impressed. In Australia, they are 12% of the total population and among them 2% vegans.  

Also just of curiosity, were there flexitarians that were not familiar with meat substitutes meaning plant-based meat varieties?  

Actually, this data collection was not done by an agency. In paragraph 3.2.1 we explain that “The data collected for this study relied on convenience sampling and snowballing” and the questionnaire was programmed in qualtrics by the researchers and placed online: https://wur.az1.qualtrics.com/jfe/form/SV_82PDxjamBNupfqB. We added a few words in 3.2.1. to clarify this lines 401-404.

Just for your information in the Dutch situation flexitarians are familiar with meat substitutes but the level of consumption varies a lot. Actually each supermarket had plant based meat replacers in it’s assortment, but the amount varies of supermarket chains. Please be aware this sample is not representative but a selection of highly involved flexitarians.  

The participants in study 2 – please add that they are females only a sit is not clear from the participants' description. I found it out at the Wrap-up 2.

We’ve added a few times more that females are overrepresented in the sample, see also response to Dora #5

Also, I will add that the article is inspired by Graça and colleagues in the introductory part, not in the discussion. In the discussion could comment on what was found to complement their study, what was different etc.

Oh, we thought to be clear about this but in response to your comment we’ve rephrased the lines 36-37.

Review and response Dora Marinova

  1. The use of quotes from Study 1 is done in a very simplistic way - it is better to incorporate excerpts in the text than just provide long lists without any obvious logic.

We don’t agree with you on this. Our logic actually is to choose for ‘running text’ on the main topics discussed in the focus groups and add some excerpts by way of illustration and give an impression of the ‘vivid’ responses and discussions. We’ve decided to leave it this way. In response to your comment we’ve gone through all the citations and shorten these were possible. In response to Diana’s comment about the letters [A, etc) we’ve repeatedly (at the first reference to a letter in each of the subsections of section 2, see lines 153, 221, 244, 282, 331) added “citation marked with...” This will help to avoid the misunderstanding by Diana that the letters refer to participants. They don’t. The letters refer to particular quotes. In addition we rechecked all citations and shortened several to have them more clear and focussed formulated.

  1. The claim about flexitarians becoming food innovators seems to be based only on consuming meat substitutes. Is this correct? If yes, this somewhat narrow perspective needs to be laid out explicitly.
  2. My biggest concern isthat the issue about availability is mentioned only in passing. The COM-B model would not work at all if meat substitutes are not available; this is obviously not the case in Holland but is still the case in many places around the world - hence, this needs to be made clear.
  3. The discussion should include some reflection about substituting meat with fruits, vegetables, pulses, nuts etc. (the good stuff) rather than processed meat substitutes.

Many thanks for these 3 comments. In response we’ve added new text: see lines 651-661.

  1. The sample in Study 2 is very much biased towards female participants. It is widely known that women are more open than men to giving up meat. This is a weakness even for an exploratory study. Such a limitation should be made explicit.

Many thanks for this comment. In response we’ve added new text: see lines 449-456.

  1. The Abstract does not state that participants in both studies were flexitarians.

Well, see lines 5 and 13-14 in which it is clearly stated that we’re dealing with flexitarians. And the title of the article seems to be also clear ;-)

  1. There needs to be a reference/s for the "intention-behaviour gap".

We’ve deleted this sentence because we didn’t feel like an obligatory reference to e.g. Vermeir & Verbeke, 2006 (although we appreciate this work and like the authors ;-)). More generally, we wanted to avoid to include too many references in this paper but have concentrated on references in which much of the relevant studies can be found, e.g. Graca-paper, Dagevos-paper .

  1. The journal uses numbers in the in-text references; however, you should not state e.g. "by [2]" but instead refer to "by Michie et al. [2]". There are several places like this. Also the original study is by Graça et al. (not Graça's study); "uner researched" should be "underresearched".

Done, thanks

  1. Can you please check the numbers in the following section: "From the remaining 483 respondents, those who indicated they do not consume meat (n=238) and do not live in the Netherlands (n= 8) were removed from further analyses. A total of 238 flexitarian participants were deemed valid for subsequent analyses."?

Thank you for noticing this typo. 237 not 238 consumers were excluded since they did not consume meat. We corrected it in the manuscript. Line 406

Review and response anonymous reviewers 2

Reviewer 2

  • The abstract needs to be written in more detail and depth
  • More explanation of mathematical methods is needed
  • Insufficient number of references

Thank you very much for reviewing our work. According to the guidelines of Sustainability the abstract should be about 200 words maximum, we do present the main messages with regard to background, methods, results and conclusion thus unfortunately there is no space to present it in the abstract in more detail.

According to your remark about the methods and the remarks of the guest editors we did do some additions and corrections in the method.

With regard to the references, as also stated above (point 7 of Dora Maronova) we have concentrated on references since we do include recent review papers which really give an up to date overview.

Round 2

Reviewer 2 Report

Accept.